# Transvaginal Ultrasound in the Diagnosis and Assessment of Endometriosis—An Overview: How, Why, and When

**DOI:** 10.3390/diagnostics12122912

**Published:** 2022-11-23

**Authors:** Angelos Daniilidis, Georgios Grigoriadis, Dimitra Dalakoura, Maurizio N. D’Alterio, Stefano Angioni, Horace Roman

**Affiliations:** 12nd University Department in Obstetrics and Gynecology, Hippokratio General Hospital, School of Medicine, Aristotle University of Thessaloniki, 546 43 Thessaloniki, Greece; 2Department of Obstetrics and Gynecology, University of Cagliari, 09042 Monserrato, CA, Italy; 3Institut Franco-Europeen Multi-Disciplinaire Endometriose (IFEMEndo), Clinique Tivoli-Ducos, 33000 Bordeaux, France; 4Faculty of Medicine, Aarhus University, 8000 Aarhus, Denmark

**Keywords:** endometriosis, ultrasound, MRI, deep infiltrating endometriosis

## Abstract

Endometriosis is a common gynaecological disease, causing symptoms such as pelvic pain and infertility. Accurate diagnosis and assessment are often challenging. Transvaginal ultrasound (TVS), along with magnetic resonance imaging (MRI), are the most common imaging modalities. In this narrative review, we present the evidence behind the role of TVS in the diagnosis and assessment of endometriosis. We recognize three forms of endometriosis: Ovarian endometriomas (OMAs) can be adequately assessed by transvaginal ultrasound. Superficial peritoneal endometriosis (SUP) is challenging to diagnose by either imaging modality. TVS, in the hands of appropriately trained clinicians, appears to be non-inferior to MRI in the diagnosis and assessment of deep infiltrating endometriosis (DIE). The IDEA consensus standardized the terminology and offered a structured approach in the assessment of endometriosis by ultrasound. TVS can be used in the non-invasive staging of endometriosis using the available classification systems (rASRM, #ENZIAN). Given its satisfactory overall diagnostic accuracy, wide availability, and low cost, it should be considered as the first-line imaging modality in the diagnosis and assessment of endometriosis. Modifications to the original ultrasound technique can be employed on a case-by-case basis. Improved training and future advances in ultrasound technology are likely to further increase its diagnostic performance.

## 1. Introduction

Endometriosis, defined as the presence of endometrial glands and stroma outside the uterus [1], is a common gynaecological disease that affects 5–10% of women of reproductive age [2], whereas in women with chronic pelvic pain or infertility, the disease frequency ranges from 35% to 50% [3]. Three distinct forms of endometriosis are widely recognized, namely ovarian endometrioma (OMA), superficial peritoneal endometriosis (SUP), and deep infiltrating endometriosis (DIE), all associated with different imaging patterns. Endometriosis, especially in most severe forms, is often coexistent with adenomyosis [4].

Accurate diagnosis and staging of endometriosis by imaging is important in order to correctly guide the clinician in the management of the disease. It is, however, challenging, as evidenced by the fact that the time interval between the first symptom and diagnosis is estimated to be 7 to 10 years [5]. The two most commonly employed imaging modalities are transvaginal sonography (TVS) and magnetic resonance imaging (MRI).

Although several studies have examined the topic of TVS in the diagnosis of endometriosis [6,7,8,9,10], its role in the accurate assessment of the disease remains debated and controversial. The aim of this narrative review is to present the available evidence in the literature regarding the role of TVS in the diagnosis and non-invasive assessment of patients with endometriosis.

## 2. Results and Discussion

### 2.1. Ovarian Endometrioma (OΜA)

Also known as “chocolate cysts” of the ovary, OMAs are present in 17–44% of patients with endometriosis [11], and are considered a marker of coexistent DIE [12]. It has been demonstrated that OMAs can be reliably diagnosed by TVS [6,13,14].

They should be carefully measured in three orthogonal planes. They typically appear as unilocular or multi-locular (less than five locules) ovarian cysts with low-level echoes (ground-glass echogenicity) and little to no vascularity (typical OMA) (Figure 1 and Figure 2). However, in about 5–10% of cases, they might appear as unilocular cysts with ground-glass echogenicity and papillary projections with little or no vascularity (atypical OMA) [15] (Figure 3). These are not true papillary projections, but fibrin and blood clots. OMAs should be described using the International Ovarian Tumor Analysis (IOTA) terminology [16]. The use of colour Doppler is helpful for differentiating an OMA from an ovarian malignancy [17].

They rarely occur in isolation and are almost always associated with other endometriotic lesions. OΜAs are often adherent to the back of the uterus or other organs, and mobility can be assessed through the use of the “sliding sign” during real-time dynamic transvaginal sonography. Adhesions may be difficult to see on ultrasound; however, the presence of free fluid may be helpful. The “pulling sleeve sign” refers to the sonographic sign of an anteverted and retroflexed uterus due to the presence of strong posterior adhesions [18]. Bilateral OΜAs in close proximity or touching each other in the pouch of Douglas (POD) are termed “kissing ovaries”. The presence of bowel endometriosis or fallopian tube endometriosis are significantly more common in women with “kissing ovaries”. Decidualisation of OΜAs may occur in pregnancy, mimicking an ovarian malignancy [19]. It should be reminded that an endometrioid or clear-cell carcinoma can develop in an OΜA [20].

### 2.2. Superficial Peritoneal Endometriosis (SUP)

Traditionally, neither TVS nor MRI have been first line tools in the diagnosis of SUP lesions. However, a novel application of TVS (saline-infusion sonoPODography) permits the direct visualization of superficial lesions with a respectable diagnostic accuracy [21]. Although larger studies are needed, it may become a useful tool in diagnosing SUP and, thus potentially reduce the number of unnecessary diagnostic laparoscopies.

### 2.3. Deep Infiltrating Endometriosis (DIE)

DIE is defined as endometriosis lesions with a depth of more than 5 mm [22]. It is considered to be the most severe form of endometriosis and is typically associated with intense clinical symptoms. There is a growing body of evidence that TVS by appropriately trained clinicians is an excellent tool in the accurate diagnosis and staging of DIE [23,24,25]. However, the most recent Cochrane meta-analysis found that ultrasound may not yet be able to replace diagnostic laparoscopy for the diagnosis of DIE [6], although it has been demonstrated that diagnostic laparoscopy may be less efficient in accurately diagnosing certain types of DIE [26].

In order to achieve higher detection rates for the diagnosis of DIE, it has been proposed to separate the pelvis during scanning in two compartments: the anterior (bladder, ureter) and the posterolateral compartment (torus uterinus, uterosacral ligaments, parametrium, vagina, rectovaginal septum, and rectosigmoid) [18].

### 2.4. The International Deep Endometriosis Analysis (IDEA) Approach

In 2016, a consensus statement was published by the International Deep Endometriosis Analysis (IDEA) group on how to systematically approach transvaginal sonography for the diagnosis and evaluation of endometriosis [18]. The authors describe four basic sonographic steps: the first step is the routine evaluation of uterus and adnexa (+sonographic signs of adenomyosis/presence or absence of endometrioma). The second step is the evaluation of transvaginal sonographic “soft markers” (i.e., site-specific tenderness and ovarian mobility). The third step is the assessment of the status of POD using a real-time ultrasound-based “sliding-sign”. Finally, the fourth step is the assessment for DIE nodules in the anterior and posterior compartments. These steps do not necessarily need to be performed in the above order, as long as all four steps are performed. The IDEA consensus has also helped in standardizing the terms and definitions used in the description of DIE in the pelvis.

Two studies have so far assessed the diagnostic accuracy of TVS in the diagnosis of DIE using the IDEA approach. The first was a prospective observational, single-centre study by Indrielle-Kelly et al. comparing the performance of TVS with that of MRI [27]: they found that, for the upper rectum DIE, TVS had sensitivity and specificity of 100% (the same as MRI) and for the sigmoid lesions, it had a sensitivity of 94% and specificity of 84% (the same as MRI). Regarding the DIE of the bladder, uterosacral ligaments, vagina, rectovaginal septum, RVS, and overall pelvis, TVS had a marginally higher specificity but lower sensitivity than MRI. The second study by Leonardi et al. was an international multicentre prospective study reporting on the performance of TVS only [28]: they found a higher TVS detection rate of DE overall than that reported by the most recent Cochrane meta-analysis (sensitivity, 79% [6]), but lower specificity (specificity, 94% [6]).

### 2.5. Anterior Compartment

#### 2.5.1. Bladder

The bladder is the most common part of the urinary tract affected by DIE [29]. The bladder base and bladder dome are the most common sites of bladder endometriosis [30], with the demarcation between the two being the uterovesical pouch [18]. DIE nodules of the bladder appear as protrusive, hypoechoic masses with or without cystic areas and regular or irregular contours (Figure 4). True DIE nodules of the bladder are those that infiltrate at least the detrusor muscle of the bladder, whereas those that are more superficial (affecting only the serosa) are not DIE nodules of the bladder. The presence of a small amount of urine in the bladder during the examination facilitates greatly in identifying DIE nodules; hence, it is the authors’ preference to assess the bladder later during TVS (and not in the beginning, when the bladder is empty). A negative “sliding-sign” between the uterus and the bladder is suggestive of bladder adhesions in the vesicouterine pouch.

#### 2.5.2. Ureter

The exact prevalence of ureteric endometriosis remains unknown, as up to 50% of cases are asymptomatic [31]. We recognise two forms: extrinsic lesions (ureter being compressed externally by the DIE lesion) and the less common intrinsic lesions (where the DIE lesion develops within the wall of the ureter) [32].

It is possible to locate the ureters by inserting the probe vaginally and following the urethra in the longitudinal plane, until its insertion in the bladder. With the probe in the anterior fornix, we move towards the pelvic sidewall on each side in order to locate the distal portion of the ureters as they enter the bladder (Figure 5). The vermiculation of the ureters helps greatly in identifying them. The presence of ureteral jets can be visualised using colour Doppler and waiting for a few seconds or minutes [30]. They can then be followed more proximally. As previously stated, it is of paramount importance to adequately assess the ureters in cases of DIE of the parametrium and uterosacral ligaments as, in such cases, the ureters are commonly infiltrated by the disease. It is the authors’ preference to routinely scan both kidneys in all patients with DIE so as to rule out possible silent hydronephrosis [33].

### 2.6. Posterior Compartment

#### 2.6.1. Torus Uterinus, Uterosacral Ligaments, Parametrium, Vagina, and Rectovaginal Septum (RVS)

In order to locate the torus uterinus (the tissue behind the cervix, in the mid-sagittal plane, between the uterosacral ligaments), we insert the vaginal probe in the anterior fornix and locate the uterus in the longitudinal section. We then draw a straight line from the level where the bladder attaches to the uterus (internal cervical os) backwards.

To locate the uterosacral ligaments, we follow a stepwise approach [34]. From the aforementioned probe position, we rotate the probe 90 degrees clockwise in order to obtain a transverse section of the cervix and then point it to the right, just lateral to the cervix, to identify the hypoechoic structures that are the uterine vessels. Just behind these, the right uterosacral ligament can be seen as hyperechoic thickening. The left uterosacral ligament can be visualised by repeating the same manoeuvre on the left side. The presence of free fluid in the POD often facilitates their visualisation in the sagittal plane (Figure 6). Endometriotic lesions of the uterosacral ligaments appear as hypoechoic thickening with regular or irregular margins within the hyperechoic stripe (Figure 7 and Figure 8). The literature suggests a wide variation in the reported accuracies of TVS for this location of DIE [35,36,37], probably owing to differences in equipment, training, and skill.

Endometriosis of the uterosacral ligament(s) should raise the suspicion of endometriosis of the parametrium and pelvic ureter that must be thoroughly assessed. The DIE of the torus uterinus appears as hypoechoic nodular thickening in the retro-cervical area, in the mid-sagittal plane (Figure 9). Parametrial endometriosis presents as a hypoechoic irregular lesion lateral to the cervical vascular plexuses.

With the probe in the posterior vaginal fornix, one can visualize forniceal endometriosis lesions as hypoechoic thickening or nodules of the vaginal wall, above the line that passes through the lower edge of the posterior lip of the cervix. The DIE in the RVS distorts the normal hyperechoic (fat-filled) layer between the vaginal and anterior rectal wall, often extending to the vagina and/or the rectum. It has been suggested that DIE in the RVS should be suspected when a hypoechoic lesion is identified between the vagina and the rectum, below the line that passes through the lower edge of the posterior lip of the cervix [38] (Figure 10 and Figure 11). “Tenderness-guided” TVS (evaluating tender sites by applying gentle pressure with the ultrasound probe) was found to have high sensitivity and specificity in the assessment of vaginal and rectovaginal DIE [39].

#### 2.6.2. Rectum and Sigmoid Colon

Bowel endometriosis affects 5 to 12% of patients with DIE [40], with the rectum and sigmoid being involved in up to 90% of those cases [41]. Depending on the depth of bowel wall infiltration (serosa, muscularis, submucosa, or even mucosa), conservative (rectal shaving or disc excision) or more radical surgical approaches (segmental bowel resection) can be employed [42].

On TVS, the rectal serosa appears hyperechoic, the muscularis hypoechoic (the longitudinal outer layer is separated from the circular inner layer by a thin hyperechoic line), the submucosa hyperechoic, and the mucosa hypoechoic. DIE nodules appear as hypoechoic lesions infiltrating the different layers of the bowel wall. Their dimensions (longitudinal, transverse, and anteroposterior) and distance from the anus should be measured. It is surgically important to correctly identify lower rectal DIE lesions as they may be more challenging to remove and may be associated with a higher risk of complications. Guerriero and Condous et al. suggested that lesions located below the level of insertion of the uterosacral ligaments are considered lower anterior rectal lesions (Figure 12), those above are upper anterior rectal lesions, those at the level of the fundus are recto-sigmoid junction lesions, whereas those above the fundus are anterior sigmoid lesions [18]. A “negative sliding sign” between the bowel and the retrocervix is strongly suggestive of obliteration of the POD secondary to DIE [36].

Although studies have suggested a good sensitivity and specificity in the diagnosis of recto-sigmoid DIE by TVS [23,43], it requires expertise [44,45] and may take significantly longer to perform compared with routine TVS [46]. Moreover, compared with MRI, it is less sensitive for upper gastrointestinal lesions (more than 16 cm from the anal margin) [47] and multifocal bowel lesions [7].

### 2.7. Other Ultrasound Techniques

As mentioned earlier, TVS for the diagnosis of DIE requires technical skills and experience. Regarding the learning curve, Guerriero et al. demonstrated that it takes an average of 17 evaluations for bladder DIE, 40 for rectosigmoid, and 44 for uterosacral ligament DIE to achieve competence [48].

Various alternatives to the routine TVS have been used to facilitate the diagnosis of DIE. Office gel sonovaginography (SVG) can be used to assess posterior compartment DIE with a high specificity and negative predictive value [49]: 20 ml of ultrasound gel are applied in the posterior vaginal fornix using a syringe, prior to insertion of the ultrasound probe, creating an “acoustic window”. The same principle of creating “an acoustic window” is observed in the office saline SVG, and a prospective study found it to be associated with a significantly higher sensitivity and specificity in the diagnosis of rectovaginal endometriosis, compared with traditional TVS, with no significantly higher patient discomfort [50]. Another approach is TVS with water contrast in the rectum (RWC-TVS) [51,52,53]: compared with traditional TVS, it may be superior for detecting infiltration of the muscularis propria [52]. Intra-operative sonorectovaginography may be helpful for assessing the posterior compartment [54]: it involves putting a catheter in the vagina and one in the rectum, filling the POD with saline via a laparoscopic port, instilling saline in the rectum via the catheter and sterile gel in the vagina following insertion of the vaginal probe. TVS with bowel preparation has been found to have a significantly higher sensitivity for detecting rectosigmoid DIE compared with diagnostic laparoscopy [26]. Rectal endoscopic sonography (RES) does not appear to be superior to TVS in the assessment of posterior compartment DIE [35,55]; however, a recent meta-analysis found RES to be more sensitive in the detection of rectosigmoid DIE [43].

### 2.8. Comparing TVS to MRI in the Assessment of DIE

Both TVS and MRI are being used in the pre-operative diagnosis and staging of DIE. We question whether there is one investigation that is superior to the other? Clinical guidance from the United Kingdom’s National Institute for Health and Care Excellence (NICE) suggests that ultrasound assessment should be considered first-line in the diagnosis of DIE [56].

Several meta-analyses have recently been published on this topic [43,57,58,59,60]. According to Guerriero et al. [58], MRI and TVS have a similar diagnostic performance in the assessment of DIE of the rectosigmoid, uterosacral ligaments, and RVS. When comparing different DIE locations, both MRI and TVS had the lowest pooled sensitivities for the detection of RVS DIE (or MRI, and the pooled sensitivity was 0.66 (95% CI, 0.51–0.79), and for TVS, the pooled sensitivity was 0.59 (95% CI, 0.26–0.86)) and the highest for rectosigmoid DIE (for MRI, pooled sensitivity of 0.85 (95% CI, 0.26–0.86) and for TVS, pooled sensitivity of 0.85 (95% CI, 0.68–0.94)). The specificities were high for both modalities in all of the aforementioned DIE locations. The quality of the included studies was considered good in most domains; however, the confidence intervals were wide and the heterogeneity was moderate to high for both modalities in most of the locations assessed.

The following meta-analyses compared the two modalities based on the location of DIE; however, for all four, there was significant heterogeneity and the studies were considered to be poor methodologically [43,57,59,60]. Regarding the DIE of the rectosigmoid, Gerges et al. found the sensitivity of TVS to be slightly better than that of MRI (for TVS, 89% (95% CI, 83–92%) versus 86% (95% CI, 79–91%) for MRI), excellent specificity for both (for TVS, 97% (95% CI, 95–98%), versus 96% (95% CI, 94–97%) for MRI); however, RES performed better than both of the modalities (sensitivity of 92% and specificity of 98%) [43]. On the same topic, Pereira et al. reported similar, high diagnostic accuracies for both modalities, with the use of bowel preparation and vaginal contrast enhancing the accuracy of MRI [60]. MRI was found to have a higher sensitivity in the diagnosis of DIE of the uterosacral ligaments, RVS, and vagina, but the specificities were similar [57]. Regarding the bladder DIE, the sensitivity of TVS was only 55% and its specificity was 99%, but a meta-analysis of the MRI data was not possible due to the small number of studies [59]. A small, retrospective study showed that MRI had a greater accuracy (96%) than TVS (92%) for bladder endometriosis [61]. Medeiros et al. performed a meta-analysis reporting a sensitivity of 64% for MRI in the diagnosis of bladder DIE [62]. MRI is considered to be superior to TVS in cases of higher bowel lesions and extra-pelvic localization of DIE [63].

### 2.9. Staging of Endometriosis

The first classification scheme for endometriosis was published by the American Fertility Society in 1979 [64], and was revised in 1996 as rASRM [65]. It remains the most commonly used classification system, and, although it generally relies on an invasive assessment of the pelvis, TVS has a high accuracy in predicting mild, moderate, and severe rASRM stages of endometriosis [66].

A new staging system, the Enzian-score [67], was developed and the #ENZIAN classification system can be used as a non-invasive and surgical description system for endometriosis [68]. It uses three compartments (A—vagina, rectovaginal space (RVS); B—uterosacral ligaments (USL)/cardinal ligaments/pelvic sidewall; C—rectum) as well as so-called F (i.e., far locations) such as the urinary bladder (FB), ureters (FU), and other extragenital lesions (FO). It additionally covers the involvement of the peritoneum (P), ovary (O), and other intestinal locations (sigmoid colon and small bowel; FI), as well as adhesions, involving the tubo-ovarian unit (T), and, optionally, tubal patency.

Several studies have assessed the performance of TVS using the #ENZIAN classification system [69,70,71]. Hudelist et al. found that TVS could detect DIE preoperatively in compartments A (vagina and RVS), B (uterosacral ligaments, parametria), C (rectosigmoid), and FB (urinary bladder), with an overall sensitivity of 84%, 91%, 92%, and 88%, respectively, and a specificity of 85%, 73%, 95%, and 99%, respectively [69]. The lesion size assessment was less precise for compartment B compared with the other compartments. The overall high accuracy of TVS in detecting and staging DIE pre-operatively was confirmed by a recent prospective multicentre study [70], as well as a retrospective single-centre study [71]. Montanari et al. found that the sensitivity of TVS ranged from 50% (#Enzian compartment FI) to 95% (#Enzian compartment A), specificity from 86% (#Enzian compartment Tleft) to 99% (#Enzian compartment FI) and 100% (#Enzian compartments FB, FU and FO), with a positive predictive value from 90% (#Enzian compartment Tright) to 100% (#Enzian compartment FO), negative predictive value from 74% (#Enzian compartment Bleft) to 99% (#Enzian compartments FB and FU), and accuracy from 88% (#Enzian compartment Bright) to 99% (#Enzian compartment FB) [70]. Di Giovanni et al. found the TVS sensitivity to be 100% for all compartments except for A and B left (97%) and FB (86%), while the TVS specificity was 100% for FB, FI, FU, and Oright; 86–98% for A, B right, C, O left, F; and 70% for B left [71].

The ultrasound-based endometriosis staging system (UBESS) was initially found to be useful for predicting the complexity of endometriosis surgery and appropriately triaging endometriosis patients to centres of excellence [14]; however, its generalizability and overall accuracy were not confirmed by later studies [72,73].

## 3. Conclusions

Accurate diagnosis and assessment of endometriosis is of paramount importance for the proper management of the disease. In this review, we presented the role of TVS and the ultrasound findings of different types and locations of endometriosis. We suggest that TVS could be the first-line imaging modality for women with suspected endometriosis as it has a good overall diagnostic accuracy for both ovarian and extra-ovarian endometriosis, it is well accepted, and it is widely available. Ultrasonography for endometriosis, however, requires specific skills, expertise, and experience that can be obtained through appropriate training. Modifications to the traditional TVS technique may further facilitate the accurate assessment of certain locations of endometriosis. The IDEA structured ultrasound approach is systematic and quite useful. TVS is already used for the staging of endometriosis using the #ENZIAN classification system with a good efficacy. Its main limitation relates to its limited ability to diagnose lesions above the rectosigmoid, multifocal bowel lesions, and extra-pelvic lesions, because of its limited field-of-view, in which case MRI is the preferred imaging modality. ΜRI should also be considered in centres where expertise in TVS are lacking and when symptoms are highly suggestive of DIE, despite a negative TVS examination. MRI should, generally, be considered a second-line investigation due to potential discomfort, bowel movements generating artefacts, higher cost, and the need for expert radiologists, with no overall clear benefit over TVS. We anticipate that raised awareness and improved, widely available ultrasound training for clinicians with an interest in the diagnosis of endometriosis, together with advances in ultrasound technology, are likely to further improve the diagnostic performance of TVS in patients with endometriosis.

## Figures and Tables

**Figure 1 diagnostics-12-02912-f001:**
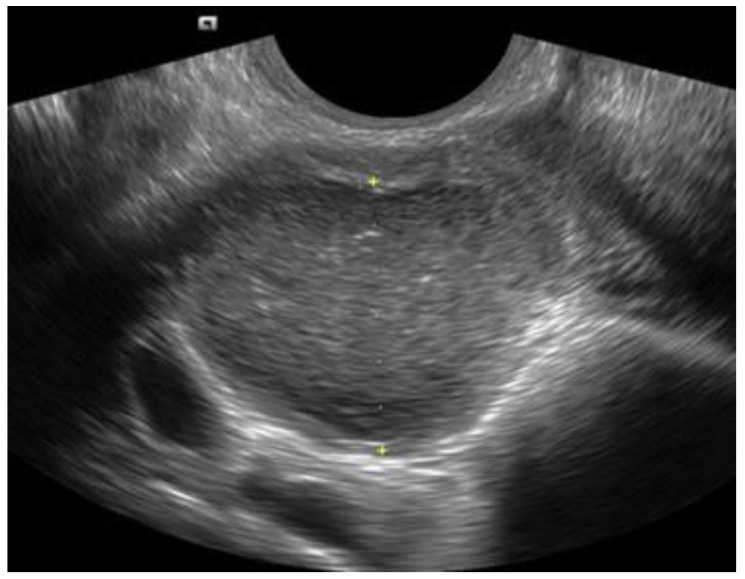
Typical ovarian endometrioma (OMA) with a ground-glass appearance.

**Figure 2 diagnostics-12-02912-f002:**
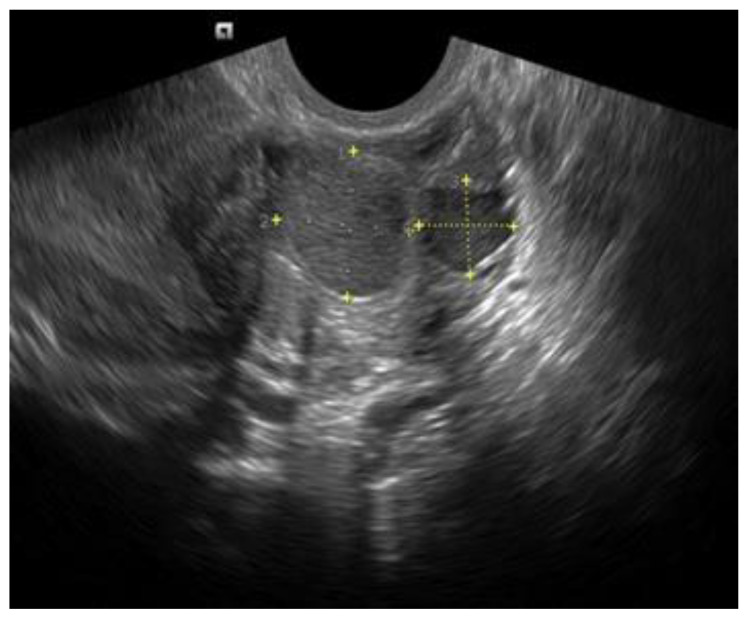
Two small OMAs with typical appearance in the same ovary.

**Figure 3 diagnostics-12-02912-f003:**
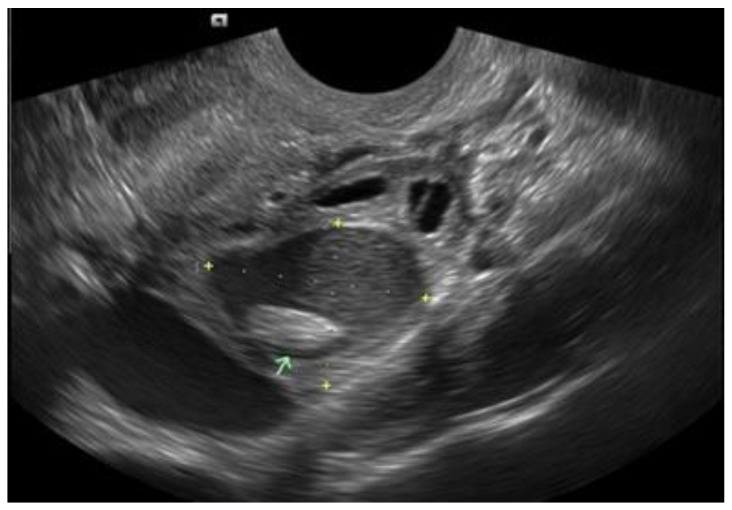
Atypical OMA with papillary projection (green arrow), likely representing a blood clot.

**Figure 4 diagnostics-12-02912-f004:**
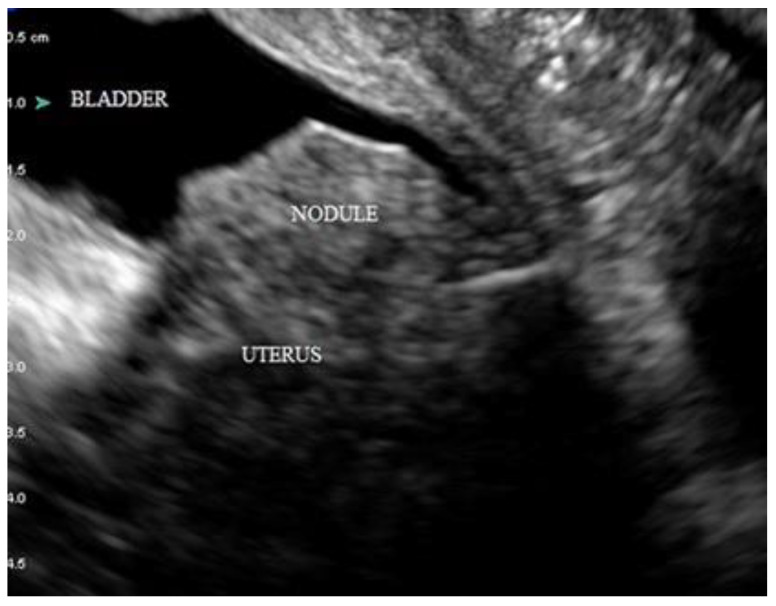
Deep infiltrating endometriosis (DIE) nodule of the bladder appearing as a protrusive nodule arising from the bladder base towards the lumen of the bladder.

**Figure 5 diagnostics-12-02912-f005:**
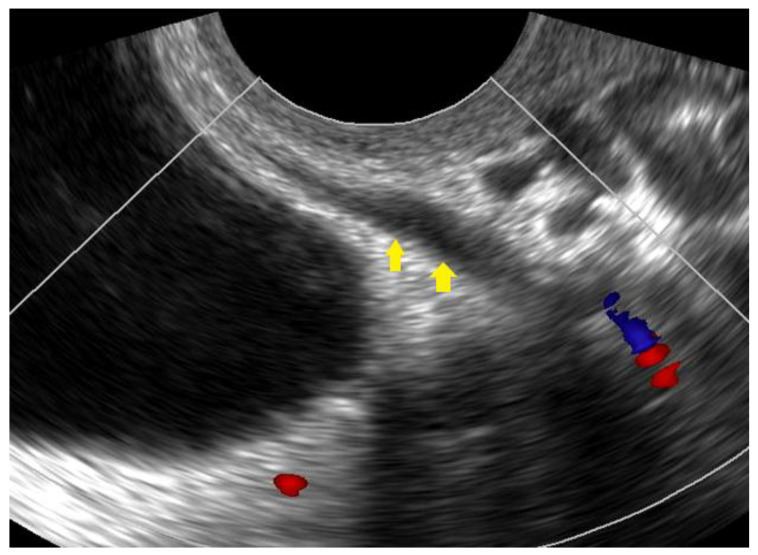
Ultrasound image showing the distal part of the ureter (yellow arrows) before it enters the bladder.

**Figure 6 diagnostics-12-02912-f006:**
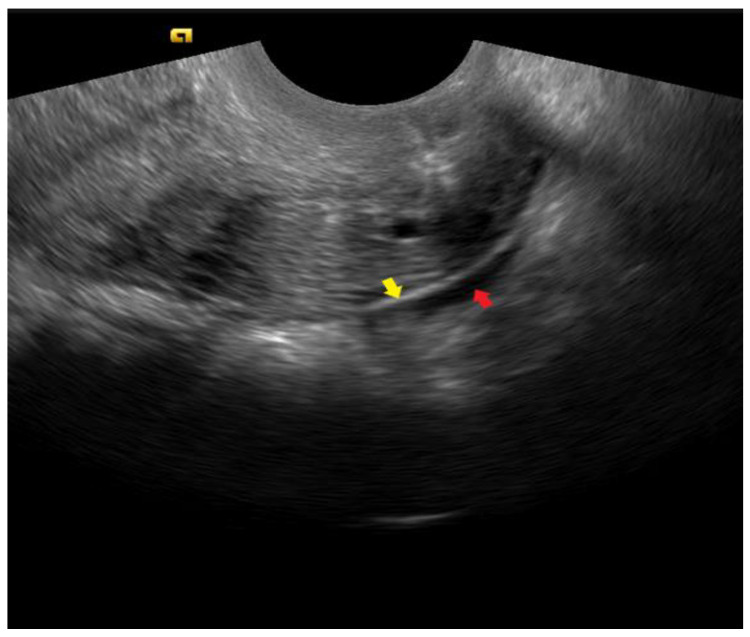
In the sagittal plane, the stretched, normal uterosacral ligament (yellow arrow) appears as a thin white line. The presence of a small amount of free fluid (red arrow) facilitates the visualization.

**Figure 7 diagnostics-12-02912-f007:**
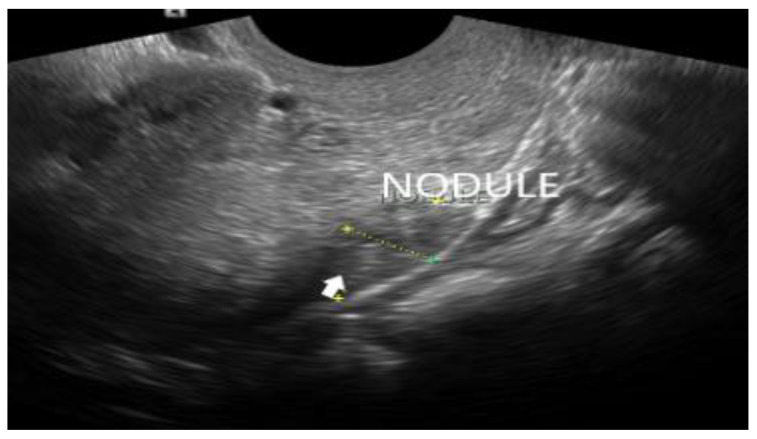
DIE nodule (white arrow) of the uterosacral ligament, appearing as a hypoechoic lesion within the white stripe.

**Figure 8 diagnostics-12-02912-f008:**
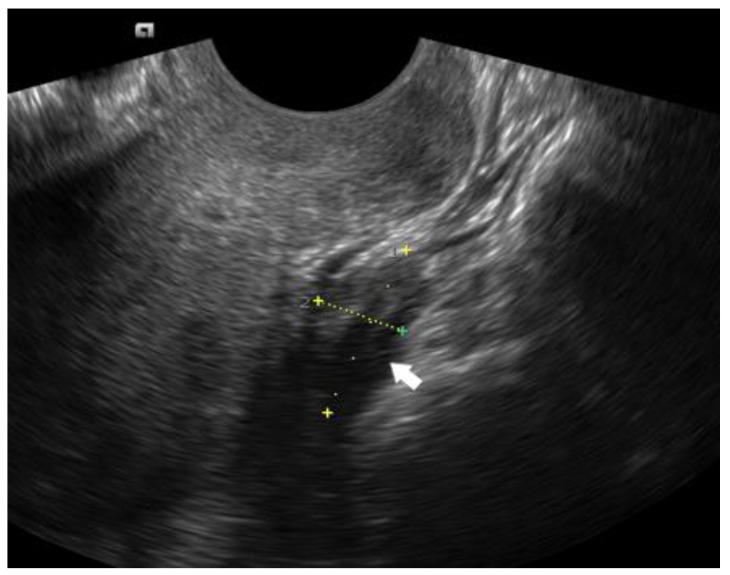
DIE nodule of the uterosacral ligament, visualized as a hypoechoic lesion (white arrow) in the sagittal plane.

**Figure 9 diagnostics-12-02912-f009:**
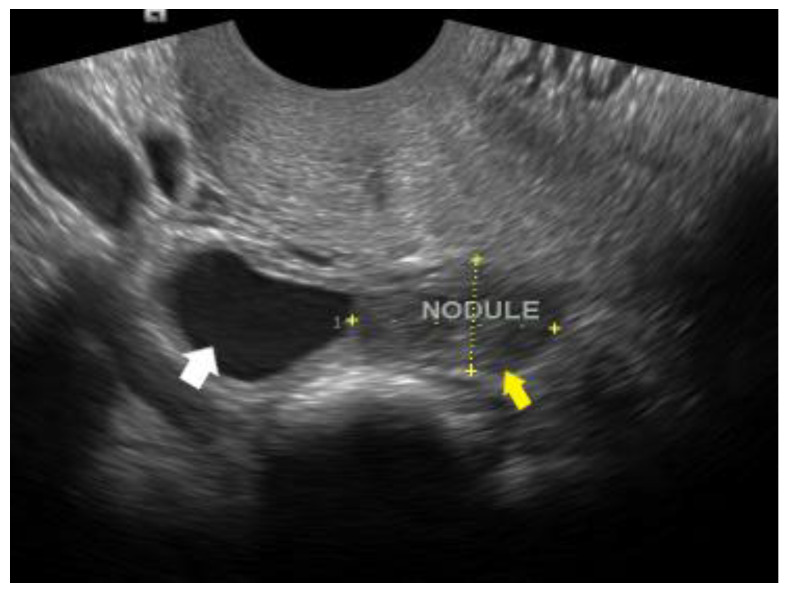
DIE nodule of the torus uterinus, appearing as a hypoechoic lesion on the rectrocervical area (yellow arrow). Adjacent to it, a hematosalpinx can be visualized (white arrow).

**Figure 10 diagnostics-12-02912-f010:**
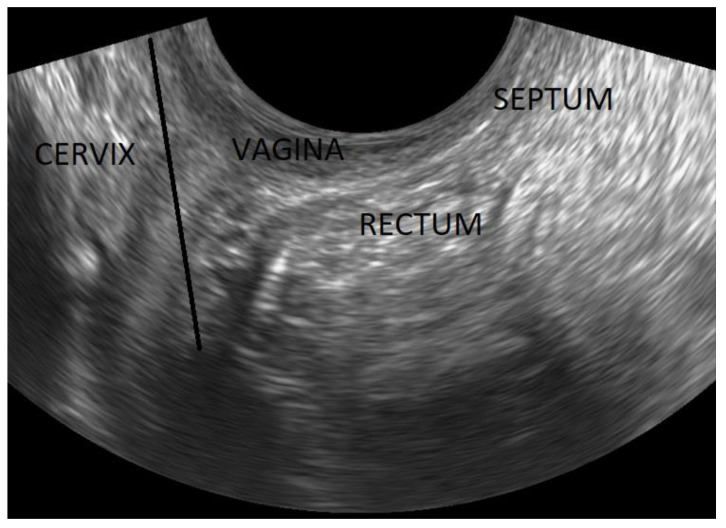
The normal rectovaginal septum as a thin white line between the posterior vaginal wall and the anterior rectal wall. The black line (which passes just below the inferior limit of the cervix) demarcates the upper limit of the rectovaginal septum.

**Figure 11 diagnostics-12-02912-f011:**
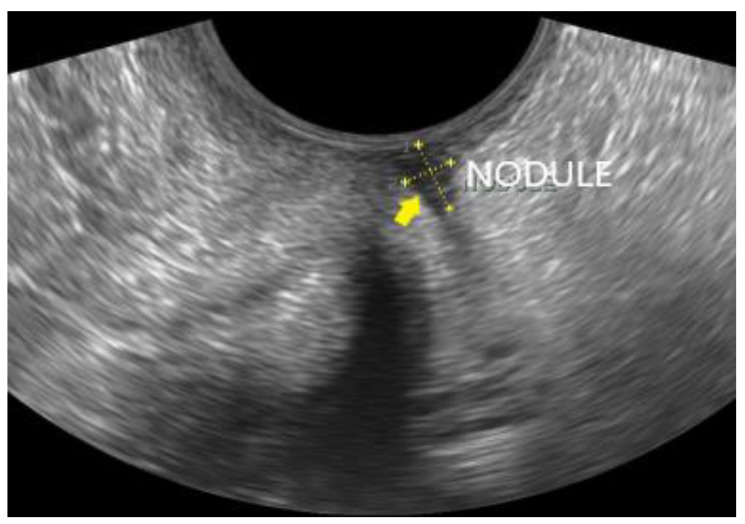
A small DIE nodule of the rectovaginal septum (yellow arrow), appearing as a hypoechoic lesion with no infiltration of the adjacent vaginal or rectal wall.

**Figure 12 diagnostics-12-02912-f012:**
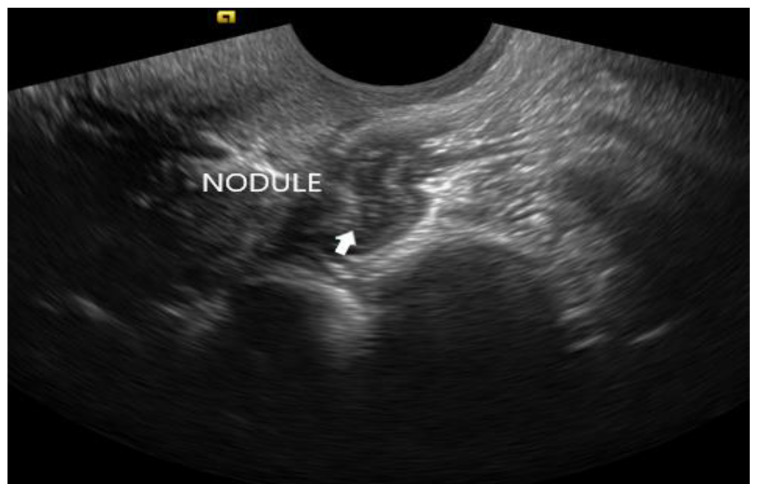
Deep endometriosis nodule of the anterior rectal wall, seen as an irregular hypoechoic lesion (white arrow).

## Data Availability

Not applicable.

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
