# Peer review of "Transvaginal Ultrasound in the Diagnosis and Assessment of Endometriosis—An Overview: How, Why, and When"

_diagnostics, 2022, doi:10.3390/diagnostics12122912_

Round 1

Reviewer 1 Report

Dear author's

I was pleased to review your article entitled "Transvaginal Ultrasound in the Diagnosis and Assessment of Endometriosis- An Overview: How, Why, When" and I have the following comments:

1. Your article is a narrative review about US diagnosis of endometriosis. There are multiple studies in the literature. Why do you chose this topic. Your article respond to ''How, Why, When"?

2. The article presents multiple US images with endometriosis lesions. I want to know your opinion. Do you think that the US examination is sufficiently for diagnosis?

3. In the section Conclusion there are not references. Please remove the references.

4. Minor English edits

Reviewer 2 Report

- This work aims to review the process of endometriosis diagnosis thanks to imaging. The Authors focused on the relevance of US images to detect different scenarios of endometriotic patients.

Endometriosis still represents one of the most challenging topics for the reproductive field yet and its diagnosis remains neglected for several years. A skilled US expert is warmly welcomed to a correct diagnosis of this disease. Even if the topic is not quite original (there are some previous works already published) the manuscript is well-written, images are generally of good quality and the readers can easily understand the message of the whole text. Some points could be improved:

1. Cite previous works regarding the imaging of endometriosis could be useful for the readers.

2. Arrows in the pictures should be reduced in their size.

3. Figure 12 should be replaced with another image with a better definition. 

Round 2

Reviewer 1 Report

Thank you for your revision.

I agree with you. Transvaginal ultrasound is sufficient in the diagnosis and non-invasive assessment of endometriosis.